# Effects of Protein Supplementation Strategy and Genotype on Milk Production and Nitrogen Utilisation Efficiency in Late-Lactation, Spring-Calving Grazing Dairy Cows

**DOI:** 10.3390/ani13040570

**Published:** 2023-02-06

**Authors:** M. J. Doran, Finbar J. Mulligan, Mary B. Lynch, Alan G. Fahey, Maria Markiewicz-Keszycka, Gaurav Rajauria, Karina M. Pierce

**Affiliations:** 1School of Agriculture and Food Science, University College Dublin Lyons Farm, W23 ENY2 Naas, Ireland; 2School of Veterinary Medicine, University College Dublin, Belfield, DO4 V1W8 Dublin, Ireland; 3Teagasc Environment Research Centre, Johnstown Castle, Y35 Y521 Wexford, Ireland

**Keywords:** dairy cow, milk production, nitrogen utilisation efficiency, late-lactation, pasture, grazing, supplementary crude protein, metabolisable protein

## Abstract

**Simple Summary:**

Challenges associated with cows producing milk for extended periods include decreased milk yield, reduced milk quality and poor nitrogen utilisation efficiency. When pasture growth is not enough to feed the dairy herd in autumn, offering dairy cows supplementary feed can negate the aforementioned challenges. Cow genetic merit can also influence milk production and cow nitrogen utilisation efficiency. However, no studies focus on lowering the overall protein of the supplementary feed and how this may impact milk production and nitrogen utilisation efficiency where cows graze outdoors. The objectives of this study were to evaluate the effects of supplementary protein concentration, cow genotype, and an interaction between supplementary protein and cow genotype on milk production and nitrogen utilisation efficiency in late-lactation, grazing dairy cows. Reducing the supplementary protein concentration from 18% to 13% decreased milk production whilst increasing faecal nitrogen. Additionally, reducing the supplementary protein concentration decreased urinary nitrogen.

**Abstract:**

The objectives of this study were to evaluate the effects of (1) protein supplementation strategy, (2) cow genotype and (3) an interaction between protein supplementation strategy and cow genotype on milk production and nitrogen (N) utilisation efficiency (milk N output/ total dietary N intake × 100; NUE) in late-lactation, spring-calving grazing dairy cows. A 2 × 2 factorial arrangement experiment, with two feeding strategies [13% (lower crude protein; LCP) and 18% CP (higher CP; HCP) supplements with equal metabolisable protein supply] offered at 3.6 kg dry matter/cow perday, and two cow genotype groups [lower milk genotype (LM) and higher milk genotype (HM)], was conducted over 53 days. Cows were offered 15 kg dry matter of grazed herbage/cow/day. Herbage intake was controlled using electric strip wires which allowed cows to graze their daily allocation-only. There was an interaction for herbage dry matter intake within cows offered HCP, where higher milk genotype (HM) cows had increased herbage dry matter intake (+0.58 kg) compared to lower milk genotype (LM) cows. Offering cows LCP decreased fat + protein yield (−110 g) compared to offering cows HCP. Offering cows LCP decreased the total feed N proportion that was recovered in the urine (−0.007 proportion units) and increased the total feed N proportion that was recovered in the faeces (+0.008 proportion units) compared to offering cows HCP. In conclusion, our study shows that reducing the supplementary CP concentration from 18% to 13% resulted in decreased milk production (−9.8%), reduced partitioning of total feed N to urine (−0.9%) and increased partitioning of total feed N to faeces (+14%) in late lactation, grazing dairy cows.

## 1. Introduction

Pasture-based dairy systems within Ireland, New Zealand and southern Australia are seasonal and aim to maximise milk production from grazed pasture [1]. Typically, the grazing season spans from February to November in the northern hemisphere [2]. Within these seasonal pasture-based systems, challenges associated with late lactation milk production include decreased milk yields, altered milk composition and poor nitrogen (N) utilisation efficiency (NUE) [3,4].

Appropriate nutrition can overcome challenges that are associated with late-lactation milk production [5,6]. Pasture-based milk production systems utilise concentrate supplement (CS) when there is a reduced availability of grazed pasture in the autumn [7]. Studies show that offering a high crude protein (CP) supplement does not result in improvements in milk production, however, it may result in an increase in harmful urinary N excretion where pasture is the basal forage in early and mid-lactation [8,9]. Therefore, based off previous research, there may be an opportunity to reduce the cows’ supplementary CP intake without impacting milk production in late lactation. This research would be beneficial considering the current European Union Nitrates Derogation whereby farmers are required to reduce supplementary CP concentration during the main grazing season. Furthermore, cow level NUE is known to decrease in late-lactation [10]. This NUE reduction can be attributed to lower dietary protein requirements as milk yields naturally decrease in late-lactation coupled with higher pasture N levels in the autumn [8]. Therefore, it may be possible to decrease the supplementary CP concentration that is offered to grazing dairy cows without impacting milk production whilst improving NUE in late lactation.

The NUE of dairy cows is an important topic because of the detrimental effect of N on the environment [11,12]. Of the two primary N excretion routes in ruminants (faecal and urinary), urinary N is the most polluting form of N excretion [13,14] but is also the most easily manipulated with differing dietary strategies [15]. Therefore, potential exists to alter the partitioning of N from urine to faeces through decreasing the supplementary CP concentration offered to cows.

The French PDI (protein digestible in the small intestine) system is based on the concept of metabolisable protein (MP) and estimates the quantity of amino acids absorbed in the small intestine from dietary protein undegraded in the rumen and from microbial protein [16]. This system can aid in more accurate balancing of dietary energy and protein requirements which results in improved dietary protein utilisation. Therefore, it is possible to reduce the supplementary CP supply whilst equalising supplementary MP supply [17]. However, knowledge of the impact of reducing the supplementary CP concentration whilst maintaining similar MP supply on milk production, N partitioning and NUE in late lactation grazing dairy cows is limited.

Cow predicted transmitting ability (PTA) values indicate the additive genetic component of a trait that a cow is expected to transmit to its offspring relative to the base population [18]. Past research completed on effects of milk production PTA has shown that higher milk production PTA cows that were offered CS had an increased milk response to CS [19] and increased dry matter intake (DMI) compared to lower milk production PTA cows [20]. In the study of Cheng et al. [21], higher Breeding Worth cows (+$198 New Zealand) were more capable of partitioning nutrients ingested from concentrates towards milk production (+1.2 kg milk yield), subsequently improving NUE (+4%), compared to lower Breeding Worth cows (+$57 New Zealand). Despite the body of published research available, there is a paucity of information on how cows of varying milk production PTA within the Economic Breeding Index (EBI) respond to differing supplement CP concentrations with equal MP.

The objectives of this study were to evaluate the effects of (1) protein supplementation strategy (PSS), (2) cow genotype and (3) an interaction between PSS and cow genotype on milk production and NUE (Milk N/N intake × 100) in late-lactation, spring-calving grazing dairy cows.

## 2. Materials and Methods

### 2.1. Animal Ethics

All procedures described in this experiment were approved by the Animal Research Ethics Committee at University College Dublin and conducted under experimental licence from the Health Products Regulatory Authority under the European Directive 2010/63/EU and S.I. No. 543 of 2012. Each person who performed procedures on experimental cows during this experiment were authorised to do so by the Health Products Regulatory Authority. Procedures conducted on the experimental cows were deemed “mild” in severity banding. Hence, no pain, suffering, or distress was observed in experimental cows, and no humane endpoints were required. This experiment was conducted at University College Dublin Lyons Farm, Celbridge, Naas, Co., Kildare, Ireland, W23 ENY2 (53°17′56.0″ N 6°32′18.0″ W).

### 2.2. Cows, Treatments and Experimental Design

A total of 36 multiparous and 12 primiparous Holstein Friesian dairy cows were selected from the spring calving herd at University College Dublin Lyons Farm. A complete randomised block design experiment, with two protein supplementation strategies (13%; (lower CP (LCP) and 18%; higher CP (HCP) CS with equal MP supply in both) offered at 3.6 kg dry matter (DM)/cow per day, and two cow genotypes (Table 1; lower milk genotype (LM) and higher milk genotype (HM)), was conducted over 53 days from 9 September to 31 October 2019. Cows were blocked on parity and balanced on days in milk (+198 ± 21.9 days), body condition score and overall EBI (within genotype groups), which is the Irish dairy total merit index [www.icbf.com (accessed on 13 June 2019)]. Cows averaged 597 kg in body weight and 2.75 lactations at the study start point (day 0). The LCP and HCP CS were formulated to contain equal PDI. When formulating the LCP, soybean meal was eliminated compared to the HCP, and lignosulfonate-treated soybean meal (SoyPass^®^) was included. As PDIE (PDI when energy is limiting) was more limiting than PDIN (PDI when N is limiting), PDIE was assumed to be the true PDI value. The PDI values were 115 g/kg for both the LCP and HCP (see Table 2). Concentrate supplementation was dispensed in the milking parlour using the Feedrite automatic system linked to cow electronic identification (Dairymaster, Kerry, Ireland) and were manufactured by Gain Feeds (Portlaoise, Ireland). Cows had *ad libitum* access to fresh water.

### 2.3. Grassland Management

Cows grazed [to 4 cm; monitored with a rising plate meter (Jenquip, Feilding, NZ)] as a single group and were offered fresh allocations of perennial ryegrass (*Lolium perenne* L.) herbage twice daily [7.5 kg DM/cow] post am and pm milking (15 kg DM per day, total). The herd moved around the grazing platform rotationally (30-day rotation), and a temporary fence was moved twice daily to provide cows with their daily allocation. Cows grazed full time (21 hours (h) per day). Pre-grazing herbage mass was determined daily before cows entered a new allocation using the “quadrant and shears” method as described by Whelan et al. [22]. The average pre-grazing herbage mass (>4 cm) was 1378 ± 285.0 kg DM/hectare (ha). Post-grazing herbage mass was also measured daily; a total of 50 measurements were taken across each grazing area using a rising plate meter (diameter 355 mm and 3.2 kg/m^2^; Jenquip, Feilding, NZ) by walking in a W-shape across the field. Post-grazing herbage mass (>4 cm) was 371 ± 191.2 kg DM/ha. The area of grassland that was allocated to cows on a daily basis depended on the pre-grazing herbage mass (Figure 1). Weekly changes in herbage quality over the experiment are shown in Figure 2.

### 2.4. Data and Sample Collection

#### 2.4.1. Herbage and Concentrate

On a daily basis, herbage samples were collected using the “quadrant and shears” method [22]. Then, on a weekly basis, daily herbage samples were pooled for chemical analyses. Concentrate samples were collected weekly for DM and then ground for analyses.

#### 2.4.2. Milk

Cows were milked twice daily at 700 h and 1500 h. Milk output was recorded, and milk sampling was facilitated using the Weighall milk metering and sampling system (Dairymaster, Kerry, Ireland). Milk samples were taken from a successive am and pm milking once per week for each cow. The ratio of am milk to pm milk used in a cow’s milk sample was equal to the ratio of the cow’s am to pm milk yield. Milk samples were collected, stored at 4 °C in a preservative (Broad Spectrum Microtabs II) and analysed once per week on the same occasion for milk composition parameters, thereby controlling any time-related confounding effects. Test day milk fat, total protein and fat + protein kg were all then determined.

#### 2.4.3. Body Weight and Body Condition Score

Individual cow body weights were measured twice daily using electronic scales as the cows exited the milking parlour through the automatic cow-drafting unit (Dairymaster, Kerry, Ireland), and then, a weekly average was calculated. All animals left the field for milking together while also receiving the same level of CS in-parlour before being weighed. This removed any bias from body weight measurements that may have occurred due to feeding. Body condition score was assessed once weekly using the method of Edmonson et al. [23] by two fully trained operators, before a consensus score was agreed upon following am milking.

#### 2.4.4. Ruminal Fluid

A sample of ruminal fluid was collected using the Flora Rumen Scoop oral oesophageal sampler (Prof-Products, Guelph, ON, Canada) once weekly following pm milking at 1600 h. To minimise saliva contamination during ruminal fluid sampling, the ruminal scoop sampling chamber was only opened after the full length of the scoop was inserted into the cow. Before removing the sampling chamber from the rumen, the sampling chamber was closed. A minimum of 8 mL of ruminal fluid was required for laboratory analysis. The sampling chamber had the capacity to collect 30 mL of ruminal fluid. The ruminal fluid pH was measured immediately (Mettler Toledo FiveGo Portable F2 pH/mV). Once collected, samples were strained through four layers of cheesecloth, a 4-mL aliquot was collected using an automatic pipette, mixed with 1 mL of 500 g/L trichloroacetic acid and cooled on ice. These were stored (−20 °C) pending analysis for ruminal fluid volatile fatty acid and NH_3_-N concentrations.

#### 2.4.5. Dry Matter Intake and Nitrogen Partitioning Study

A N partitioning study was conducted during week 2 of the experiment (206 ± 21.9 days in milk). Herbage DMI and cow NUE were estimated over a period of 6 days. Herbage DMI was determined using the n-alkane technique of Dove and Mayes [24]. Cows were dosed with a paper bolus impregnated with 500 mg of the n-alkane n-dotriacontane, for a period of 12 days following am and pm milking. From days 7 to 12, samples of the CS, herbage, milk and faeces were all collected in the am (0600–0900 h) and pm (1400–1700 h). Faecal samples were collected from 0600 h to 0700 h before am milking when cows naturally defaecated, and if not, samples were collected per rectum post am milking at 0800 h. All faecal samples were then placed in a forced-air oven at 55 °C for 72 h. Samples of milk were collected during am (0700 h) and pm (1500 h) milking each of the 6 days for each cow.

### 2.5. Sample Analyses

#### 2.5.1. Herbage, Concentrate and Faecal Sample Analysis

Herbage, CS and faecal samples were dried in a forced air oven at 55 °C and were ground in a hammer mill fitted with a 1-mm screen (Lab Mill, Christy Turner, Ltd., Ipswich, UK). The DM content of samples was determined after drying for 16 h at 105 °C [25]. The ash content was determined following combustion in a muffle furnace (Nabertherm GmbH, Lilienthal, De) at 550 °C for 5.5 h [26]. The N content of samples was determined by combustion on LECO and CP content calculated (N × 6.25; FP 528 Analyzer, Leco Corp., St. Joseph, MI, USA) [27]. The ether extract of herbage and CS samples was determined using Soxtec instruments (Tecator, Höganäs, SE) and light petroleum ether. Neutral detergent fibre of herbage and CS samples were determined using the sodium sulfite method of Van Soest et al. [28], adapted for use in the ANKOM^TM^ 220 Fiber Analyzer (ANKOM^TM^ Technology, Macedon, NY, USA). Acid detergent fibre was determined using the method of Van Soest et al. [28]. For the NDF method, a thermostable α-amylase and 20 g of Na2S was included. Starch content of feed samples was analyzed using the Megazyme Total Starch Assay Procedure (product no: K-TSTA; Megazyme International Ireland Ltd., Wicklow, Ireland). A sample of 100 mg was required, and reagents used in the procedure included: sodium acetate buffer (100 mM, pH 5.0) plus calcium chloride (5 mM); sodium acetate buffer (200 mM, pH 4.5) plus calcium chloride (5 mM); sodium acetate buffer (600 mM, pH 3.8) plus calcium chloride (5 mM); sodium hydroxide solution (1.7 M); and MOPS buffer (50 mM, pH 7) plus calcium chloride (5 mM) and sodium azide (0.02% *w*/*v*). The concentration of water-soluble carbohydrates was determined as described by Dubois et al. [29].

#### 2.5.2. Milk Analysis

Concentrations of milk fat, protein, lactose, casein, milk urea N (MUN) and somatic cell count were determined in a commercial milk laboratory (National Milk Laboratories Ltd., Wolverhampton, UK) using mid-infrared spectrometry (Milkoscan FT6000, Foss Analytical A/S, Hillerod, DK) [30].

#### 2.5.3. Ruminal Fluid Analysis

Ruminal fluid was allowed to thaw in the refrigerator overnight for 16 h at 4 °C and was then centrifuged at 2100× *g* for 10 minutes (min) at 4 °C. Then, 1 mL of supernatant was diluted 1 in 5 with distilled H_2_O and then centrifuged at 1600× *g* for 15 min at 4 °C. Next, 200 μL of supernatant was combined with three reagents and used to determine ruminal fluid NH_3_-N concentration using a spectrophotometer (UVmini-1240, Shimadzu Corp., Kyoto, Japan). Ruminal fluid was prepared for volatile fatty acid analysis by mixing 250 μL of ruminal fluid with 3.75 mL of distilled H_2_O; to this, 1 mL of internal standard solution (0.5 g 3-methylvaleric acid in 1000 mL of 0.15 M oxalic acid) was added. The resulting solution was centrifuged at 1600× *g* and filtered through a syringe-tip filter (PTFE, 22-mm diameter, 0.45 μm) into 2-mL gas chromatography vials. Concentrations of volatile fatty acid were determined using a Scion 456-GC (Scion Instruments, Scotland, UK) fitted with a DB-FFAP capillary column (15 m × 0.53 mm; 1.00 μm, Agilent Technologies Ireland Ltd., Cork, Ireland). Volatile fatty acids were separated using programmed oven temperatures: the temperature was initially set to 45 °C for 2 min, then raised at a rate of 10 °C per min to 170 °C and held for 30 seconds; in the next phase, the temperature was raised by 40 °C per min up to 220 °C and held for 2 min. Injector and detector temperature was 240 °C. Spitless injection was used. Hydrogen and air were used as the carrier gas and N was used as the make-up gas. Methyl-valeric acid was used for the internal standard and volatile fatty acids were identified based on retention time of standards. Peak areas were integrated using Compass CDS software (version 2.0).

#### 2.5.4. Dry Matter Intake and Nitrogen Partitioning Analysis

Herbage DMI was determined by extracting n-alkanes from herbage, CS and faecal samples according to the method of Dove and Mayes [24]. Following extraction, samples were analysed for concentrations of n-alkanes (C31, C32, C33 and C34) by gas chromatography using a Scion 456-GC (Scion Instruments, Scotland, UK) fitted with a 30-m capillary column with an internal diameter of 0.53 mm coated with 1.5 μm of dimethyl polysiloxane (Agilent Technologies Ireland Ltd., Ireland Ltd., Cork, Ireland). These data were then applied to the following modified equation to calculate herbage DMI (HDMI)/cow per day [31]: HDMI=[(Fi/Fj)(Dj+IcCj)−IcCi]/[Hi−(Fi/FjHj)], where Fi and Fj are the concentrations of naturally occurring odd-chain (feed derived) and even-chain (dosed n-dotriacontane n-alkane in faeces, respectively (mg/kg); Hi and Hj are the concentrations of natural odd-chain and even-chain n-alkanes in herbage, respectively (mg/kg); Dj is the daily dose rate of the even-chain n-alkanes (mg/kg); Ic is the daily concentrate intake (kg/d); and Ci and Cj are the concentrations of natural odd-chain and even-chain n-alkanes in concentrate feed (mg/kg), respectively. These data were used to calculate N partitioning according to Whelan et al. [15] as follows: N intake (kg/day)=[(HDMI (kg/day)× kg N/kg of DM herbage)+(kg of CS DMI × kg of N/kg of DM concentrate)];Faecal N (kg)=(kg of faecal DM excretion × kg of N/kg of DM faeces); Milk N =(kg of milk yield × kg of N/kg milk) and estimated urinary N (kg)=(N intake (kg) – faecal N (kg) – Milk N (kg)). Faecal N (kg) was determined by: 1000 mg C32 (C32 dosed per day)/C32 concentration recovered in the faeces [24]. The CP concentration of pasture grazed by cows was 24.4% during the 6-day nitrogen partitioning study.

### 2.6. Statistical Analysis

Residuals of data were checked for normality and homogeneity of variance by histograms, QQ-plots and formal statistical tests as part of the UNIVARIATE procedure of SAS (version 9.4, SAS Institute Inc., Cary, NC, USA). Somatic cell count data were not normally distributed and were transformed by raising the variable to the power of lambda. The appropriate lambda value was obtained by conducting a Box-Cox transformation analysis using the TRANSREG procedure of SAS [32]. The transformed somatic cell count data were used to calculate *p*-values. The corresponding least squares means and standard error of the mean of the non-transformed somatic cell count data are presented in the results for clarity. The relationships between total feed N intake, milk N output, estimated urinary N excretion and faecal N excretion were tested for linear associations using Pearsons Correlation Coefficient. If associations were significant, the coefficient of determination was generated using the REG procedure of SAS. Milk production and composition, ruminal fermentation, body condition score and N excretion parameters were analysed using repeated measures ANOVA (MIXED procedure). The fixed effects in the model were milk genotype (LM vs. HM), PSS (LCP vs. HCP), week and their interaction. Block was the random factor and cow was the most suitable subject and was considered the experimental unit. Week of experiment was the repeated unit. The overall model was as follows:Yijk=μ+τj+xk+τXjk+εijk
where

Y*ijk* = *i*th cow in the *j*th treatment of the kth week;*τj* = fixed effect of treatment, where *j* = genotype (LM or HM) and PSS (LCP or HCP);*xk* = the fixed effect of week, where *k* = week 1–8;*τ*X*jk* = the interaction of the *j*th treatment with the *k*th week;and *εijk* = residual error.

Milk production and composition and ruminal fermentation parameter measurements were taken before experimental cows were assigned to treatments for use as covariates in the statistical model where appropriate. Heterogenous compound symmetry, unstructured, autoregressive, heterogeneous 1st order autoregressive, Toeplitz and heterogenous Toeplitz were (co)variance structures considered. The model with the lowest Bayesian Information Criterion value was selected. A Tukey adjustment was used for multiple comparisons. Probabilities of *p* < 0.05, *p* < 0.01 and *p* < 0.001 were selected as significance levels. Where “NS” is displayed in *p*-value columns, this highlights non-significance (*p* > 0.05).

## 3. Results

Where no PSS × genotype interaction was observed for measured parameters, the results only focus on main effects. However, as treatments were in a factorial arrangement, the potential interaction between PSS and genotype was left in the statistical model.

### 3.1. Dry Matter Intake, Milk Production and Milk Composition (Table 3)

Cows offered HCP had decreased HDMI (−1.12 kg; *p* < 0.001) and total DMI (TDMI; −1.12 kg; *p* < 0.001) compared to cows offered LCP. There was a PSS × genotype interaction (*p* < 0.01) observed for both HDMI and TDMI. The LM cows offered HCP had a decreased HDMI and TDMI compared to the HM cows offered HCP (*p* < 0.05) and the LM cows offered LCP (*p* < 0.001), whilst the LM cows offered LCP did not differ from HM cows offered LCP (*p* > 0.05). The HM cows offered LCP had increased (*p* < 0.05) HDMI and TDMI compared to the HM cows offered HCP.

**Table 3 animals-13-00570-t003:** Effects of protein supplementation strategy (PSS) genotype on dry matter intake, milk production and milk composition ^1^.

Genotype	Lower Milk Genotype		Higher Milk Genotype			Significance		
PSS	LCP (n = 12)	HCP (n = 12)	LCP (n = 12)	HCP (n = 12)	SEM	Genotype	PSS	Interaction
Dry matter intake								
Herbage, kg/d	15.67 ^a^	14.11 ^b^	15.37 ^a^	14.69 ^c^	0.148	NS	<0.001	<0.01
Total, kg/d	19.27 ^a^	17.71 ^b^	18.97 ^a^	18.29 ^c^	0.148	NS	<0.001	<0.01
Milk production, kg/d								
Milk yield	17.20	18.40	17.78	20.35	0.690	NS	<0.01	NS
Fat	0.83	0.87	0.84	0.89	0.028	NS	NS	NS
Protein	0.70	0.75	0.71	0.79	0.023	NS	<0.01	NS
Casein	0.54	0.59	0.56	0.63	0.018	NS	<0.01	NS
Fat + protein	1.52	1.62	1.55	1.68	0.049	NS	<0.05	NS
Lactose	0.74	0.79	0.76	0.87	0.030	NS	<0.01	NS
Milk composition, %								
Fat	4.78	4.69	4.70	4.57	0.092	NS	NS	NS
Protein	4.05	4.11	4.02	4.04	0.043	NS	NS	NS
Casein	3.20	3.26	3.17	3.20	0.038	NS	NS	NS
Lactose	4.28	4.29	4.25	4.31	0.022	NS	NS	NS
MUN, mg/dL milk ^2^	31	32	29	32	0.9	NS	<0.05	NS
SCC, × 10^3^ cells/mL ^3^	88	74	81	71	10.9	NS	NS	NS

^1^ Genotype × week, PSS × week and week *p*-values: milk yield = NS, NS and *p* < 0.01, respectively; fat kg = NS, NS and *p* < 0.01, respectively; protein kg = *p* < 0.05, NS and *p* < 0.01, respectively; casein kg = NS, NS and *p* < 0.01, respectively; fat + protein kg = NS, NS and *p* < 0.01, respectively; lactose kg = NS, NS and *p* < 0.01, respectively; fat % = NS, NS and *p* < 0.01, respectively; protein % = *p* < 0.05, NS and *p* < 0.01, respectively; casein % = *p* < 0.05, NS and *p* < 0.01, respectively; lactose % = NS, NS and *p* < 0.01, respectively; MUN = NS, *p* < 0.05 and *p* < 0.01, respectively; and SCC = NS, NS and *p* < 0.01, respectively. SED for lower and higher milk genotype cows is as follows: Herbage dry matter intake = 0.148; Total dry matter intake = 0.148; Milk yield = 0.691; Fat kg = 0.027; Protein kg = 0.228; Casein kg = 0.018; Fat + protein kg = 0.048; Lactose kg = 0.030; Fat % = 0.092; Protein % = 0.035; Casein % = 0.038; Lactose % = 0.022; MUN % = 0.9; and SCC = 8.2. SED for lower and higher crude protein treatments is as follows: Herbage dry matter intake = 0.148; Total dry matter intake = 0.148; Milk yield = 0.690; Fat kg = 0.027; Protein kg = 0.228; Casein kg = 0.018; Fat + protein kg = 0.048; Lactose kg = 0.030; Fat % = 0.092; Protein % = 0.035; Casein % = 0.038; Lactose % = 0.022; MUN % = 0.9; and SCC = 8.3. SED for PSS × genotype is as follows: Herbage dry matter intake = 0.210; Total dry matter intake = 0.210; Milk yield = 0.976; Fat kg = 0.039; Protein kg = 0.032; Casein kg = 0.025; Fat + protein kg= 0.069; Lactose kg = 0.042; Fat % = 0.130; Protein % = 0.050; Casein % = 0.054; Lactose % = 0.031; MUN % = 1.3; and SCC = 11.6. Dry matter intake data corresponds to week 2 only, whilst milk production and composition data corresponds to weeks 1–8. ^2^ MUN = milk urea nitrogen. ^3^ SCC = somatic cell count. ^a–c^ means with different superscripts within the same row, differ (*p* < 0.05).

Cows offered HCP had increased yields of milk (+1.89 kg; *p* < 0.01), protein (+0.07 kg; *p* < 0.01), casein (+0.06 kg; *p* < 0.01), fat + protein (+0.11 kg; *p* < 0.05) and lactose (+0.08 kg; *p* < 0.01) compared to cows offered LCP. However, genotype did not affect any milk production parameter (*p* > 0.05).

Cows offered HCP had increased MUN concentration (+2 mg/dL milk; *p* < 0.05) compared to cows offered LCP. Genotype had no effect on any milk composition parameter (*p* > 0.05).

Cows offered HCP had increased (*p* < 0.05)) milk fat yield compared to cows offered LCP during week 8 (Figure 3) whilst cows offered HCP had increased (*p* < 0.05) milk protein yield compared to cows offered LCP during weeks 1–3 and 5–8 of the study (Figure 4).

### 3.2. Ruminal Fermentation (Table 4)

There was no effect of PSS or genotype on ruminal pH or NH_3_-N concentration (*p* > 0.05). However, there was a PSS × genotype interaction (*p* < 0.05) observed for ruminal valeric concentration. The HM cows offered HCP had an increased ruminal valeric concentration compared to the HM cows offered LCP (*p* < 0.01) and the LM cows offered HCP (*p* < 0.01), whilst the LM cows offered HCP did not differ from the LM cows offered LCP (*p* > 0.05). Cows offered the HCP had a decreased ruminal acetic to propionic concentration ratio (−0.35 units; *p* < 0.001) but an increased ruminal propionic concentration (+2.15 mmol/L; *p* < 0.01) compared to cows offered LCP. The HM cows had a decreased ruminal acetic to propionic concentration ratio (−0.16 units; *p* < 0.05) but an increased ruminal propionic concentration (+1.76 mmol/L; *p* < 0.05) compared to the LM cows.

**Table 4 animals-13-00570-t004:** Effects of protein supplementation strategy (PSS) and genotype and on ruminal fermentation parameters.

Genotype	Lower Milk Genotype		Higher Milk Genotype			Significance		
PSS	LCP (n = 12)	HCP (n = 12)	LCP (n = 12)	HCP (n = 12)	SEM	Genotype	PSS	Interaction
Ruminal pH	6.66	6.62	6.64	6.55	0.046	NS	NS	NS
Ruminal NH_3_-N_,_ mmol/L	4.76	5.09	5.01	5.28	0.222	NS	NS	NS
VFA ^1^								
Total VFA, mmol/L	111.21	113.07	114.52	118.41	4.301	NS	NS	NS
Acetic: propionic	3.7	3.33	3.52	3.19	0.062	<0.05	<0.001	NS
Acetic, mmol/L	72.81	71.92	74.21	74.30	2.755	NS	NS	NS
Propionic, mmol/L	19.85	21.35	20.96	23.76	0.857	<0.05	<0.01	NS
Butyric, mmol/L	14.48	15.00	15.22	15.85	0.548	NS	NS	NS
Isobutyric, mmol/L	1.13	1.09	1.18	1.18	0.041	NS	NS	NS
Valeric, mmol/L	1.20 ^a^	1.27 ^a^	1.24 ^a^	1.54 ^b^	0.055	<0.01	<0.01	<0.05
Isovaleric, mmol/L	1.50	1.51	1.50	1.65	0.092	NS	NS	NS

^1^ VFA = volatile fatty acids. ^ab^ means with different superscripts within the same row, differ (*p* < 0.05).

### 3.3. Nitrogen Partitioning (Table 5)

Cows offered HCP had decreased total feed N excreted in the faeces (−0.006 kg; *p* < 0.01) and increased total feed N excreted in the urine (+0.005 kg; *p* < 0.05) compared to cows offered LCP. Furthermore, cows offered HCP had a decreased proportion of total feed N excreted in the faeces (−0.008 proportion units; *p* < 0.01) and an increased proportion of total feed N excreted in the urine (+0.007 proportion units; *p* < 0.05) compared to cows offered LCP. There was a PSS × genotype interaction (*p* < 0.01) observed for total feed N intake whereby the HM cows offered HCP did not differ from the HM cows offered LCP (*p* > 0.05). However, the LM cows offered HCP had decreased total feed N intake (*p* < 0.01) compared to the LM cows offered LCP. The HM cows offered HCP had increased total feed N intake (*p* < 0.05) compared to the LM cows offered HCP, whilst the HM cows offered LCP did not differ from the LM cows offered LCP (*p* > 0.05). Cows that were offered HCP had a decreased body weight (*p* < 0.001) compared to cows that were offered LCP.

**Table 5 animals-13-00570-t005:** Effects of protein supplementation strategy (PSS) and genotype on nitrogen (N) partitioning.

Genotype	Lower Milk Genotype		Higher Milk Genotype			Significance		
PSS	LCP (n = 12)	HCP (n = 12)	LCP (n = 12)	HCP (n = 12)	SEM	Genotype	PSS	Interaction
Intake, kg/d								
Total feed N	0.689 ^a^	0.658 ^b^	0.678 ^ac^	0.681 ^ac^	0.0058	NS	<0.05	<0.01
Cow body weight, kg	644	590	618	575	12.9	NS	<0.001	NS
Milk yield, kg/day	19.45	21.30	20.35	22.19	0.919	NS	NS	NS
N excreted, kg/d								
Milk	0.131	0.131	0.132	0.132	0.0008	NS	NS	NS
Faeces	0.042	0.034	0.041	0.036	0.0023	NS	<0.01	NS
Urine	0.504	0.509	0.503	0.508	0.0027	NS	<0.05	NS
N excreted as a proportion of total feed N intake, % ^1^								
Milk	0.195	0.195	0.196	0.197	0.0015	NS	NS	NS
Faeces	0.061	0.052	0.061	0.053	0.0032	NS	<0.01	NS
Urine	0.744	0.751	0.743	0.750	0.0042	NS	<0.05	NS
N excreted, % ^2^	80.9	80.5	80.4	80.3	0.15	NS	NS	NS

^1^ N proportions = N out [faeces, urine, milk (kg/d)]/N intake (kg/d). ^2^ N excreted = N out [(faeces + urine output (kg/d))/N intake (kg/d)] × 100. ^–^ means with different superscripts within the same row, differ (*p* < 0.05).

## 4. Discussion

A supplementation level of 3.6 kg DM/cow per day is common during autumn within grazing systems in Ireland as pasture growth is reduced at this time of year [7]. Furthermore, there has been an increasing focus placed on decreasing the supplement CP concentration within grazing systems in light of the current European Union Nitrates Derogation [33].

### 4.1. DMI, Milk Production and Milk Composition

Past research has shown that decreasing the supplement CP concentration does not impact HDMI or TDMI [8,34]. Contrary to these studies, cows offered LCP had increased HDMI and TDMI in our study and is consistent with the study of Doran et al. [35]. Cows were offered 3.1 kg DM CS/cow per day which had either 140 or 180 g CP/kg CS DM in the aforementioned study. Cows in our experiment were offered 3.6 kg DM/cow per day which had either 130 or 180 g CP/kg CS DM. Research shows that where dietary NDF concentration increases, intake and digestibility decreases [36]. However, as all cows grazed the same pasture, dietary NDF concentration differences between cows offered LCP and HCP were presumed to be minimal in our findings. Furthermore, cows in the study of Doran et al. [35] also had an increased HDMI when offered LCP. The authors attributed this result to a numerical increase in cow BW. During our N partitioning study, cows that were offered LCP had an increased BW (+54 kg) compared to cows that were offered HCP and this result was responsible for the increased HDMI. It is unlikely that increased HDMI was responsible for the difference in cow BW at the end of the N partitioning study as Vazquez and Smith [37] attributed variation in DMI to cow BW, and not vice versa.

Within cows offered HCP, HM cows had increased HDMI, whereas within cows offered LCP, there was no difference in herbage intake. Diets that are higher in CP concentration usually have a lower NDF concentration, thus, both digestibility and therefore, intake potential can be increased [37]. As all cows grazed the same herbage simultaneously, and 80–81% of the cows’ diet consisted of grazed herbage, we would not have expected differences in herbage DMI of the overall diets. Research shows that where cows of differing genetic merit are compared, higher genetic merit cows have greater HDMI capacity [38]. Therefore, HM cows had an increased DMI capacity compared to LM cows where the HCP supplement was offered.

Despite the differences in HDMI and TDMI, cows that were offered LCP had decreased yields of milk, fat + protein and lactose. These results were unforeseen, as the CS constituted only a small proportion of the overall diet (19.7–20.3% on a DM basis); therefore, we would not have expected differences in milk production through reducing the supplemental CP concentration by 50 g CP/kg CS. Furthermore, the authors focused on equalising MP supply between both supplements used in this study. The PDI concept focuses MP which is important for microbial protein synthesis and AA absorption in the small intestine. The PDI concept considers feeds that have differing RDP and RUP values and how these feeds will be utilized in cows’ digestive system. Ruminal energy supply that is available to utilise this protein efficiently is also considered. Ensuring that 100% of MP is supplied can translate to increased milk production [17] compared to diets that are deficient in MP. Therefore, looking at dietary MP supply rather than just CP supply is a more biologically sensible measure of dietary protein for ruminants. When equalising MP supply, regular soybean (65% RDP and 35% RUP) [39] inclusion was eliminated in LCP and was replaced by SoyPass^®^ (26% RDP and 74% RUP) [39]. This allowed the current authors to decrease the supplementary CP concentration without reducing the intestinal availability of methionine and lysine; two of the most limiting essential amino acids for milk production [40]. The decreased milk protein yield in cows that were offered LCP was due to the decreased RDP:RUP through SoyPass^®^ inclusion in LCP and is similar to the findings of Savari et al. [41]. However, good quality herbage is a rich RDP source (70–90% RDP and 10–30% RUP) [42] and since herbage composed 81% of the cows’ diet, we would not have expected a reduction in supplemental RDP to negatively impact yields of milk protein. These results are relevant to the Irish and European dairy industry given recent restrictions around supplementing [33]. Further research should investigate if differing RDP: RUP in low CP CS have an impact on milk production where cows are pasture-based and are in late-lactation.

It is well-established that an increase in dietary CP concentration can equate to an increase in MUN, indicating an excess of dietary CP in dairy cows [43,44,45]. Offering cows HCP increased MUN concentration in our findings and concurs with Burke et al. [9]. Despite this finding, milk fat + protein yield was still increased with HCP, indicating that the extra dietary CP intake was not all in excess of the cows’ requirements. Previous autumn grazing studies conducted at this research facility show that herbage CP levels have ranged from 236 to261 g CP/kg DM [4,46]. Herbage CP concentration was lower in our study, averaging 201 g CP/kg DM. Despite this result, overall dietary CP concentrations for all treatments in our study were above the optimum threshold recommended by Katongole and Yan [47] for maximum milk yield.

Numerous studies have demonstrated the effect of dairy cow genotype on milk production [48,49,50]. Doran et al. [46] found that HM cows had increased milk yield compared to LM cows where herbage was the basal forage and where cows were greater than 200 days in milk. Milk yield or milk fat + protein yield did not increase in HM cows compared to LM cows in our findings. Higher milk genotype cows were in the national top 20% for milk yield PTA, whereas the LM cows placed in the bottom 20% in our study. Therefore, for us to have observed a difference in milk yield, greater differences in milk yield PTA between genotypes were required.

### 4.2. Ruminal Fermentation

Offering cows HCP increased ruminal propionic concentration compared to offering cows LCP. Consequently, the ruminal acetic to propionic concentration ratio was decreased with HCP in this study. Cows that were offered HCP also had increased yields of milk and lactose. Ruminal propionic acid is a precursor of glucose, which is a precursor of milk lactose, the main osmotic regulator of milk yield [51,52]. Furthermore, the ruminal valeric concentration and valeric proportion were increased in cows offered HCP. Increasing this short chain fatty acid is important for improving microbial fermentation and can have beneficial effects on milk production [53].

### 4.3. Nitrogen Partitioning

When using the method of Whelan et al. [15] to calculate N partitioning, an important assumption was that negligible levels of N were accreted in cow body tissue. Considering that cow body weight did not change from the start to the end of that experiment, this was a valid assumption. This was an important assumption as it enabled the authors to account for all dietary N intake that was fed. As cow body weight did not change from the start to the end of our experiment (Figure 5), authors felt that this assumption also applied to our experiment. However, the N partitioning study was conducted only at the beginning of the trial (week 2) and in this short period, cow body weight changed (increased in LCP cows and decreased in HCP cows, probably as a consequence of differences in milk production and energy balance).

Previous studies show that increasing the supplement CP increases total feed N intake [34,45]. Contrary to the above studies, results of our experiment show that where cows were offered HCP, total feed N intake was decreased, and this result was because cows that were offered HCP also had a decreased HDMI. Furthermore, cows that were offered HCP had decreased faecal N excretion and increased estimated urinary N excretion. This result may have occurred because of the increase in supplementary CP and the simultaneous increase in MUN. However, N intake was increased where cows were offered LCP during the N partitioning study (+14 g N); therefore, MUN should be also have been increased in cows offered LCP. However, an important caveat to this study was that N intake was only measured for one week, as N partitioning studies in grazing dairy cows are expensive and laborious. If N partitioning was measured for the entire experiment, the result may have been different. Meanwhile, offering cows HCP did not increase milk N output. This result occurred as the additional dietary N intake was partitioned to urine. These results are consistent with Whelan et al. [15] and Burke et al. [9] where increasing supplement CP concentration increased estimated urinary N excretion in early and mid-lactation, respectively. In our findings, total feed N intake and estimated urinary N excretion were poorly correlated, where total feed N intake only explained 14% of the variation in urinary N excretion and is in contrast with Mulligan et al. [8]. Furthermore, as cows offered LCP in our N partitioning study had increased body weight, these cows were likely in positive energy balance (EB). It has been documented that plasma urea concentration is known to decrease during periods of improved EB [54]; therefore, improving EB can aid in decreasing plasma urea N and urinary N excretion. If EB remained similar to cows offered HCP, the estimated urinary N excretion difference would have been greater between treatments. Thus, it is important to note that urinary N excretion may have been influenced by dietary CP inclusion and/or EB in this study.

Contrary to what was expected, offering cows LCP did not increase cow NUE as there was no increase in the proportion of total feed N recovered in the milk. Dietary protein was oversupplied in both LCP and HCP cows, primarily due to the high pasture CP content (24.4%). Cows offered HCP responded to the soybean meal inclusion by increasing milk production. Soybean meal is a rich source of lysine [55]; therefore, dietary lysine may have been marginally deficient in cows offered the LCP diet and this could have impacted ruminal fermentation, microbial protein synthesis and milk protein output. Further research should investigate if (1) differing supplemental lysine concentrations impact milk production where cows are offered methionine-rich pasture in late-lactation and (2) differing cow genotypes respond differently to these supplementation strategies. Despite the above result, cow level NUE remained high for the stage of lactation (averaging 19.6%) compared to the study of McKay et al. [4] (averaging 13.6%) and Reid et al. [56] (averaging 13.2%). Cows in the study of McKay et al. [4] had an average total feed N intake of 0.786 kg and a milk N output of 0.107 kg, whilst cows in our study had a lower total feed N intake (0.677 kg) and an increased milk N output (0.132 kg). The cow level NUE differences mentioned herein was due to a combination of differing N intake and cow genetic merit. Alternatively, cow NUE differences between studies could be due to important pasture nutrient interactions such as the ratio of pasture water-soluble carbohydrates to CP [57,58]. As pasture chemical composition varied across time, it is possible that cow NUE also has varied across time in our findings.

There were no effects of genotype on N partitioning or cow level NUE in our findings, contrary to the New Zealand study of Cheng et al. [21] where cows were also in late lactation (+200 days in milk) and grazed autumn herbage. Our study investigated differences in NUE within the Irish dairy genetic merit indexing system (EBI) with a particular focus on the milk production sub-index (€54 versus €33), whereas the study of Cheng et al. [21] investigated differences across the New Zealand genetic merit indexing system (Breeding Worth). Cows in our study all had a similar overall EBI (€154), whereas cows in the aforementioned study had different Breeding Worth values ($198 versus $57). Therefore, for us to have observed differences in NUE with respect to cow genotype, cows needed to differ in their overall EBI also.

## 5. Conclusions

Despite supplying equal MP through both supplements, cows offered LCP had decreased yields of milk, fat + protein and lactose in late lactation. However, there was no effect of PSS on cow NUE. Genotype had no effects on milk production or cow NUE. Cows that differed in genetic make-up did not respond differently to altering the supplement CP concentration. Cows offered LCP had lower MUN concentration and urinary N excretion, which was likely caused by dietary crude protein concentration and energy balance. Furthermore, total feed N intake recovered in the urine was reduced, whilst total feed N intake recovered in the faeces was increased where cows were offered LCP. In summary, offering cows LCP decreased milk production, maintained NUE and reduced urinary N excretion compared to offering cows HCP in late lactation, grazing dairy cows.

## Figures and Tables

**Figure 1 animals-13-00570-f001:**
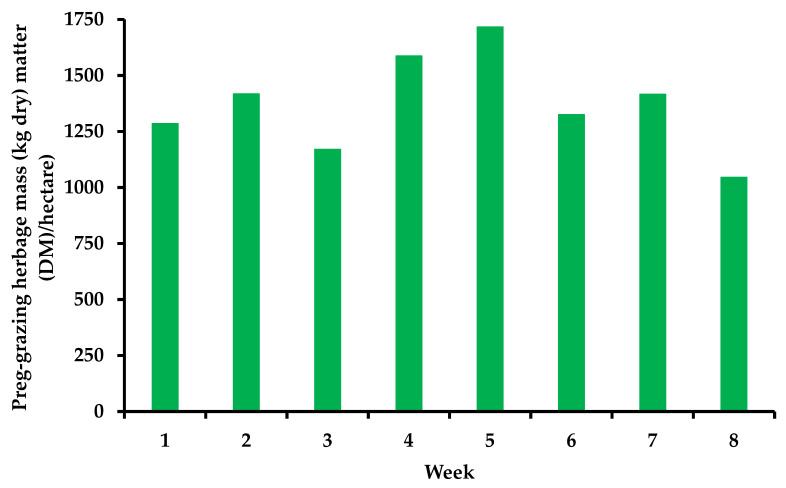
Changes in pre-grazing herbage mass on a weekly basis.

**Figure 2 animals-13-00570-f002:**
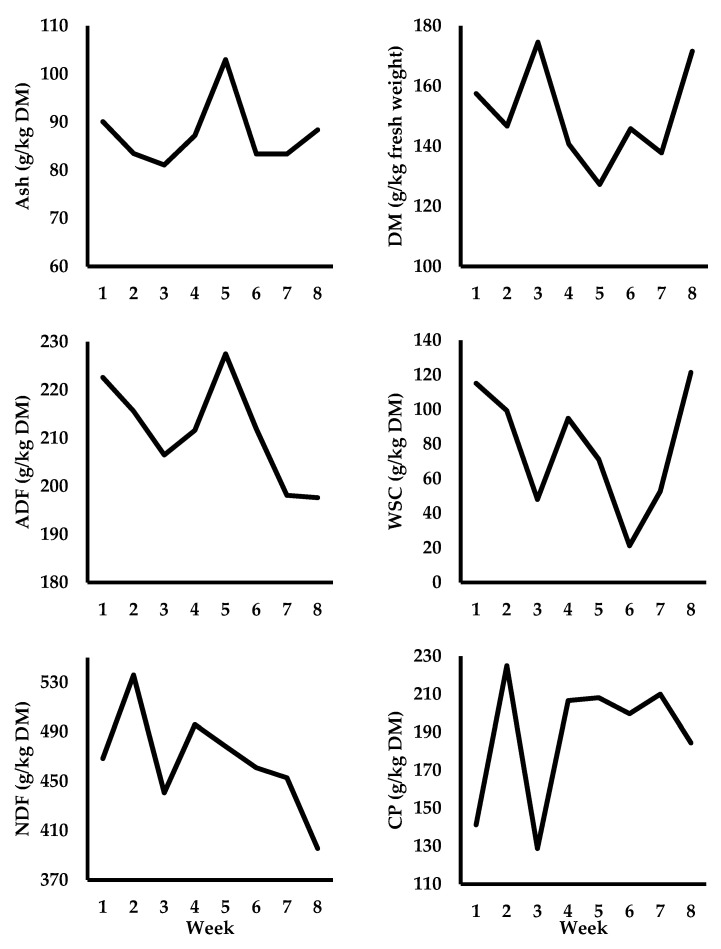
Changes in autumn herbage quality offered to dairy cows during the experiment (week 1 = September 9 and week 8 = October 28). A nitrogen partitioning study was conducted during week 2. Standard deviations across the 8-week experimental period were: ±6.5 g ash/kg DM; ±15.4 g DM/kg of FW (fresh weight); ±10.0 g ADF/kg DM; ±33.2 g WSC/kg DM; ±38.3 g NDF/kg DM; and ±32.5 g CP (crude protein)/kg DM.

**Figure 3 animals-13-00570-f003:**
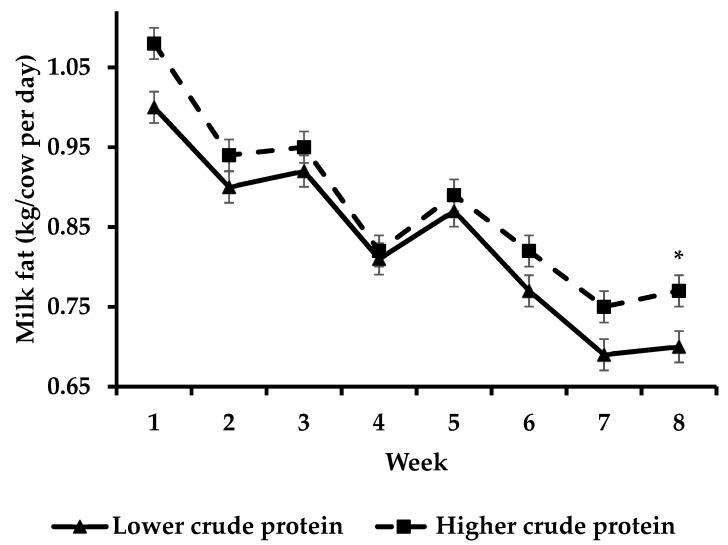
Effect of protein supplementation strategy × week on milk fat yield [asterisks (*) indicate a significant difference for week].

**Figure 4 animals-13-00570-f004:**
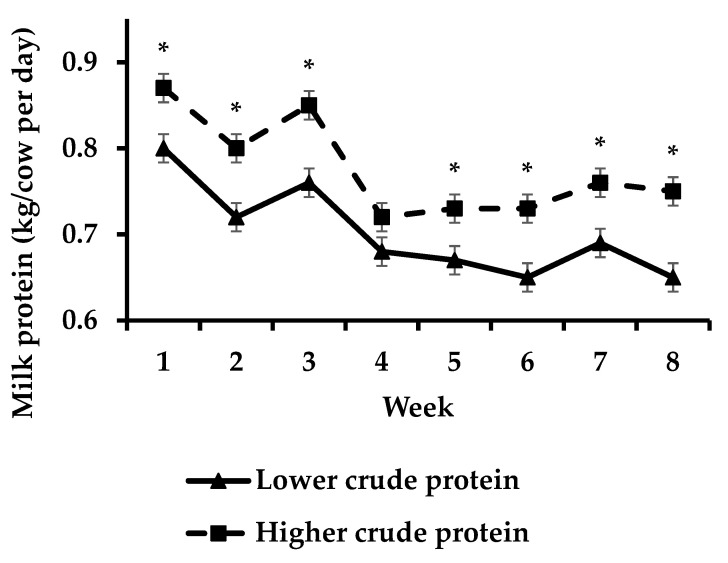
Effect of protein supplementation strategy × week on milk protein yield [asterisks (*) indicate a significant difference for week].

**Figure 5 animals-13-00570-f005:**
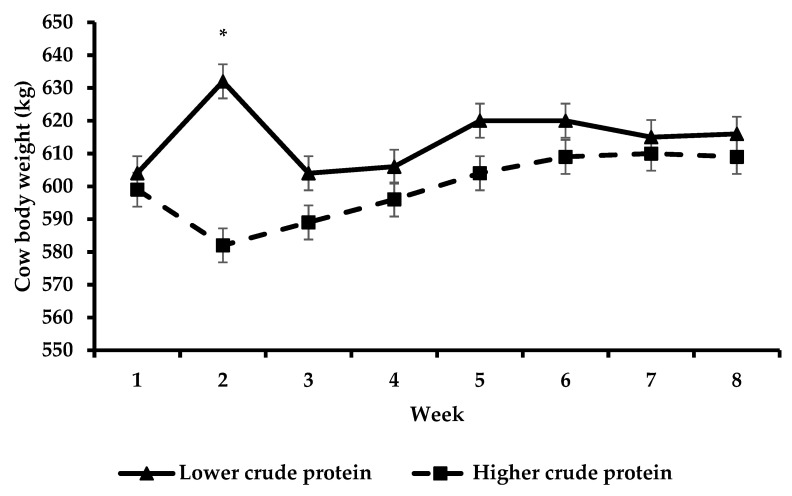
Body weight change over time with respect to lower crude protein and higher crude protein treatments. The effect of week was non-significant (asterisks (*) indicate a significant difference for week).

**Table 1 animals-13-00570-t001:** Genotype profile of cows in the experiment ^1^.

Item	Lower Milk Genotype	National Percentile	Higher Milk Genotype	National Percentile
Genetic parameter				
Milk kg	−60	Bottom 20%	130	Top 20%
Fat kg	6	Top 40%	10	Top 5%
Protein kg	3	Bottom 40%	7	Top 10%
Fat %	0.13	Top 5%	0.08	Top 20%
Protein %	0.08	Top 5%	0.06	Top 10%

^1^ National percentiles apply to the year 2019 [www.icbf.com (accessed on 13 June 2019)].

**Table 2 animals-13-00570-t002:** Chemical composition, ingredient inclusion level and metabolisable protein content of experimental feedstuffs.

Item	LCP ^1^	HCP ^2^	Herbage
Chemical composition, g/kg dry matter unless stated ^3^			
Dry matter, g/kg	866	869	156
Ash	124	84	88
Crude protein	137	188	201
Neutral detergent fibre	212	192	498
Acid detergent fibre	136	109	223
Water soluble carbohydrates	-	-	83
Ether extract	29	35	42
Starch	260	269	-
Ingredient inclusion level of concentrates, g/kg ^4^			
Barley	221	225	
Maize	234	231	
Maize distiller grain with solubles	-	100	
Sugar beet pulp pellets 8 mm	227	40	
Soyabean meal	-	220	
Soya hulls	50	50	
Soyabean oil	9	9	
SoyPass^® 5^	125	-	
Palm oil blend	6	6	
Monocalcium diphosphate	23	16	
Sugarcane molasses	45	45	
Calcium carbonate	13	16	
Sodium chloride	18	18	
Magnesium oxide	16	15	
Gain cattle premix ^6^	6	6	
Metabolisable protein, g/kg ^7^			
PDIE ^8^	115	115	85
PDIN ^9^	94	128	151
PDIA ^10^	61	63	54

^1^ LCP = lower crude protein concentrate supplement of 13%. ^2^ HCP = higher crude protein concentrate supplement of 18%. ^3^ DM = dry matter. ^4^ All grains were ground. ^5^ SoyPass^®^ = lignosulfonate-treated soybean meal. ^6^ Gain cattle premix consisted of the following: 0.1 g copper/kg for LCP and HCP; 15 g calcium/kg and 14.3 g calcium/kg for LCP and HCP, respectively; 7.5 g phosphorus/kg for LCP and HCP; 7.5 g sodium/kg and 7.4 g sodium/kg for LCP and HCP, respectively; 9.5 g magnesium/kg and 9.4 g magnesium/kg for LCP and HCP, respectively; 10,000 IU vitamin A/kg for LCP and HCP, respectively; 2500 IU vitamin D/kg for LCP and HCP, respectively; and 62.5 IU vitamin E/kg for LCP and HCP, respectively. ^7^ The French PDI system details metabolisable protein. ^8^ PDIE = protein digestible in the small intestine (PDI) when energy is limiting. ^9^ PDIN = PDI when nitrogen limiting. ^10^ PDIA = PDI when protein undegraded in the rumen but digestible in small intestine [17].

## Data Availability

The data presented in this study are available on request from the corresponding author.

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
