# Peer review of "Effects of Protein Supplementation Strategy and Genotype on Milk Production and Nitrogen Utilisation Efficiency in Late-Lactation, Spring-Calving Grazing Dairy Cows"

_animals, 2023, doi:10.3390/ani13040570_

Round 1
Reviewer 1 Report
Dear authors, it seems to me that this manuscript has relevance in the scientific world. However, many points affect the quality of the manuscript.
General comments:
I'm not sure, however I think the manuscript needs some language corrections considering the type of English (American or British) stated in the author's instruction manual.
Dear authors, in some parts of the text you refer to grazing cows. For me, this is ad libitum feeding. At other points, it is written (eg lines 412-413) that the animals receive an exact amount of food. It is one thing to write grazing animals and another to write animals fed with diets containing grass and concentrate.
Abstract:
The abstract should be improved by adding more information. Remember, the abstract is the document that first represents your manuscript.
Introduction:
The introduction needs to be improved. I recommend rewriting it by adding numbers and considering protein and genotype as focuses, as written in the title and aim of the study. In the current presentation, it is more focused on environmental contamination.
Discussion:
The topic of discussion is to explain chemically and biologically what happened to get those results and on this topic I can only see speculation, reviews and comparative studies. Improve it.
Conclusion
This conclusion needs to be improved.
Specific comments:
Lines 12-19: These lines have no correlation with the objective of the study. In this sense, removing these lines is my suggestion and write something else related to the objective of the study.
Lines 22-24: Rewrite these lines; you can be more objective here.
Lines 24-25: if this is a conclusion? This has no correlation with the objective. Rewrite it.
Line 28: What does NUE mean? Fully describe the initials the first time they appear, here and throughout the text.
Line 31: Wait......, he offered 15 kg DM? Weren't the cows in a grazing system?
Line 32: What does HM mean?
Line 33: What does LM mean?
Lines 36-39: Ok, this is a possible conclusion. However, this conclusion tells me nothing because that result was already expected. Improve it.
Line 53: How much is HIGH? These values may be different depending on the reader.
Lines 53-54: I recommend removing these lines because this is not an inefficient use of the protein. This is probably why the N use efficiency is not ideal.
Lines 55-73: Perhaps this is part of the focus of the manuscript; however, the writing style here does not show that. This text here is not correctly correlated with the study; however the information is not wrong, just rewrite it following another approach. My suggestion is based on the fact that the objective of the study is not environmental contamination. Rewrite it.
Lines 80-82: So, I understand that this is the differential of your manuscript. Low amount of studies on this topic? Perhaps at the local geographic level there are few studies, but the protein metabolism and the efficiency of its use are known worldwide.
Lines 83-87: I think this is independent of race; I think this is more of a metabolic situation. Of course, in general terms. I mean supplementation. The PTA is clearly a genetic statement. Lines 87-92 try to explain these genetic differences; however, I think the writing style may be different. Perhaps adding numbers on genotypes or production level and nitrogen efficiency could make this topic more explicit and objective.
Lines 119-120: Add the average body weight and age of the animals.
Line 121: How much concentrate supplement was offered to the animals?
Table 2: I am concerned about the differences in the ingredients and contents of PDIN. Those changes can influence the results and lead to biased results.
Line 189: Was there a fasting period before weighing the animals at any time? The variation in the feeding of the animals can influence the weight of the animals.
Line 225: Is better to add the number of hours.
Line 230: Are you sure the name "Soxtex" is correct?
Line 275: I'm confused about the repeatability of some subtitles. I recommend combining this subtopic with the 2.4.5 subtopic.
Line 326: These types of lines can be avoided. It is repetitive with the table title. Remove similar lines throughout the text.
Lines 410-412: Wait. Here you are referring to the results found in the article by Doran et al, right? If yes, the word "Cows" (line 410) must begin with a lowercase letter.
Lines 408-428: Ok, I read and understood the content of the text. However, what was the effect that influenced their results? What is the explanation? Why did you get those results? Rewrite it.
Lines 430-433: Wait what? Are you sure you describe the CS, in the amount supplemented, as a small proportion of the total diet?
Lines 433-436: It is right; however, the focus of his work is not energy. You need to explain this part better. Rewrite it.
Line 473: Propionic is a precursor for glucose and glucose for lactose. Rewrite it.
Lines 473-478: I don't understand how these lines improve your manuscript if they don't explain your results. Rewrite it.
Lines 481-485: Are you sure that this thought is correct? I think the explanation is different. Rewrite it.
Lines 486-523: Rewrite it following the previous recommendations.
Lines 525-531: Those lines need to be improved. This conclusion is more of a description of results. The conclusion must be correlated with the objective of the study.
Author Response
General comments
Reviewer 1: I'm not sure, however I think the manuscript needs some language corrections considering the type of English (American or British) stated in the author's instruction manual.
AU: The authors have proofed the entire document and language corrections have been applied throughout.
Reviewer 1: Dear authors, in some parts of the text you refer to grazing cows. For me, this is ad libitum feeding. At other points, it is written (eg lines 412-413) that the animals receive an exact amount of food. It is one thing to write grazing animals and another to write animals fed with diets containing grass and concentrate.
AU: The authors thank the Reviewer for their point here and feel that the terminology around this is something to consider. Cows grazed outdoors during the study but were allocated an exact amount of grazed grass per animal (15kg DM). This was possible due to to measuring pre-grazing yield per hectare with a plate meter and using a strip wire to allocate grass.
Abstract:
Reviewer 1: The abstract should be improved by adding more information. Remember, the abstract is the document that first represents your manuscript.
AU: The authors were conscious of the word limit for the journal; however, some more information has been added.
Introduction:
Reviewer 1: The introduction needs to be improved. I recommend rewriting it by adding numbers and considering protein and genotype as focuses, as written in the title and aim of the study. In the current presentation, it is more focused on environmental contamination.
AU: Changes made throughout introduction.
Discussion:
Reviewer 1: The topic of discussion is to explain chemically and biologically what happened to get those results and on this topic I can only see speculation, reviews and comparative studies. Improve it.
AU: Changes have been made.
Conclusion
This conclusion needs to be improved.
AU: Conclusion has been amended.
Specific comments (Amended text is highlighted in green in the revised manuscript).
Reviewer: Lines 12-19: These lines have no correlation with the objective of the study. In this sense, removing these lines is my suggestion and write something else related to the objective of the study.
AU: The authors are aiming to put some context into place for the study to the lay person in lines 12-19 but do admit that wording and phrasing can be improved. Please see revisions. Lines 12-23.
Reviewer 1: Lines 22-24: Rewrite these lines; you can be more objective here.
AU: Rewritten. Now lines 21-23.
Reviewer 1: Lines 24-25: if this is a conclusion? This has no correlation with the objective. Rewrite it.
AU: The authors think it's best to remove this line. Removed.
Reviewer 1: Line 28: What does NUE mean? Fully describe the initials the first time they appear, here and throughout the text.
AU: NUE now fully described. Line 26-27.
Reviewer 1: Line 31: Wait......, he offered 15 kg DM? Weren't the cows in a grazing system?
AU: Yes, the cows were in a grazing system. However, strip wires were moved ahead of cows constantly, so cows only had access to enough grazed grass to get them to the next milking (15 kg DM/ day, total). More information has been added here. Lines 31-32.
Reviewer 1: Line 32: What does HM mean?
AU: Clarified. Line 34.
Line 33: What does LM mean?
AU: Clarified. Line 35.
Reviewer 1: Lines 36-39: Ok, this is a possible conclusion. However, this conclusion tells me nothing because that result was already expected. Improve it.
AU: Have added some more information here . Lines 38-41.
Line 53: How much is HIGH? These values may be different depending on the reader.
AU: It was suggested that we remove this sentence anyway.
Reviewer 1: Lines 53-54: I recommend removing these lines because this is not an inefficient use of the protein. This is probably why the N use efficiency is not ideal.
AU: Now removed. Line 54.
Lines 55-73: Perhaps this is part of the focus of the manuscript; however, the writing style here does not show that. This text here is not correctly correlated with the study; however the information is not wrong, just rewrite it following another approach. My suggestion is based on the fact that the objective of the study is not environmental contamination. Rewrite it.
AU: The authors accept that the primary aim of the study is on milk production and NUE. Therefore, the section has been amended. However, route of N excretion is an important topic of the study and is one that we were able to determine. Therefore, the authors feel that it is appropriate to leave lines 68-72. Lines 54-81 have been amended.
Lines 80-82: So, I understand that this is the differential of your manuscript. Low amount of studies on this topic? Perhaps at the local geographic level there are few studies, but the protein metabolism and the efficiency of its use are known worldwide.
AU: The authors accept that nitrogen metabolism is indeed a widely researched topic. However, no studies have been conducted that reduce the supplement CP whilst equalising MP supply. Within a grazing system. No studies focus on the EBI and the milk sub index of the EBI and how these cows might respond to the different supplementary CP concentrations.
Lines 83-87: I think this is independent of race; I think this is more of a metabolic situation. Of course, in general terms. I mean supplementation. The PTA is clearly a genetic statement. Lines 87-92 try to explain these genetic differences; however, I think the writing style may be different. Perhaps adding numbers on genotypes or production level and nitrogen efficiency could make this topic more explicit and objective.
AU: Numbers on the above now added. Lines 86-90.
Lines 119-120: Add the average body weight and age of the animals.
AU: Info. now added Lines 120-121.
Line 121: How much concentrate supplement was offered to the animals?
AU: This information has already been mentioned. Lines 116.
Reviewer 1: Table 2: I am concerned about the differences in the ingredients and contents of PDIN. Those changes can influence the results and lead to biased results.
AU: The authors acknowledge the Reviewers point here. Where concentrate supplements are going to differ in crude protein, ingredient inclusion levels will also differ. The focus of the experiment was on dietary metabolisable protein. PDIN was supplied in excess with respect to treatment diets. Therefore, to ensure that 100% of the cows MP requirement was met, we authors had to shift our attention to ensuring that PDIE was sufficient and equal. PDIE is equalised between the two supplements, therefore removing potential bias with respect to overall MP supply. However, the fact that differences in milk production due to more RDP in the 18% supplement has been given some consideration in the discussion. Lines 456-461.
Reviewer 1: Line 189: Was there a fasting period before weighing the animals at any time? The variation in the feeding of the animals can influence the weight of the animals. Have added clarity on this. Line191-194.
AU: There was no fasting period. Have added clarity on this. Lines 191-194.
Reviewer 1: Line 225: Is better to add the number of hours.
AU: Have now added number of hours-only. "Overnight" removed. Line 229.
Reviewer 1: Line 230: Are you sure the name "Soxtex" is correct?
AU: Should read "Soxtec". Now amended. Line 234.
Reviewer 1: Line 275: I'm confused about the repeatability of some subtitles. I recommend combining this subtopic with the 2.4.5 subtopic.
AU: The authors have laid out the sections so that 2.4 explains how the data was collected. 2.5 explains how the samples were analysed. The authors feel it is important that these sections are kept separate. However, if the Reviewer still feels strongly about this, the authors can amend.
Reviewer 1: Line 326: These types of lines can be avoided. It is repetitive with the table title. Remove similar lines throughout the text.
AU: All of these lines are now removed. Line 341, etc.
Reviewer 1: Lines 410-412: Wait. Here you are referring to the results found in the article by Doran et al, right? If yes, the word "Cows" (line 410) must begin with a lowercase letter.
AU: The authors wish to start a new sentence here; hence, why "Cows" has gotten a capital letter.
Reviewer 1: Lines 408-428: Ok, I read and understood the content of the text. However, what was the effect that influenced their results? What is the explanation? Why did you get those results? Rewrite it.
AU: Rewritten. Now lines 417-431.
Reviewer 1: Lines 430-433: Wait what? Are you sure you describe the CS, in the amount supplemented, as a small proportion of the total diet?
AU: Yes, it constituted a small proportion of the overall diet; hence why we would not have expected negative effects of lowering the supplementary CP on milk production. Lines 442-446.
Reviewer 1: Lines 433-436: It is right; however, the focus of his work is not energy. You need to explain this part better. Rewrite it.
AU: Rewritten. Now lines 447-454.
Reviewer 1: Line 473: Propionic is a precursor for glucose and glucose for lactose. Rewrite it.
AU: Amended. Now line 491-492.
Reviewer 1: Lines 473-478: I don't understand how these lines improve your manuscript if they don't explain your results. Rewrite it.
AU: The authors also feel that this doesn't add much to the discussion. Part of this paragraph has now been deleted. Line 492-493.
Reviewer 1: Lines 481-485: Are you sure that this thought is correct? I think the explanation is different. Rewrite it.
AU: The authors have reconsidered this piece of text and have rewritten as recommended. Lines 498-503.
Reviewer 1: Lines 486-523: Rewrite it following the previous recommendations.
AU: Rewritten. Now lines 504-542.
Reviewer 1: Lines 525-531: Those lines need to be improved. This conclusion is more of a description of results. The conclusion must be correlated with the objective of the study.
AU: The authors have amended this section. ie. 1) effect of protein supplementation strategy on milk production and NUE, 2) effect of genotype on milk production and NUE, and 3) potential interaction between PSS and genotype on milk production and NUE.
Reviewer 2 Report
Dear Editor and Authors,
I send you my review about the article Effects of protein supplementation strategy and genotype on milk production and nitrogen utilisation efficiency in late-lactation, spring-calving grazing dairy cows”.
The scope of the paper, as reported in the aim was to evaluate the effects of protein supplementation strategy, cow genotype, and their interaction on milk production.
In my opinion the article result original, and enough well structured. However, it need, before to be published, of some minor revisions that I report below.
General comments:
In general, acronyms can help the reader better understand the article. However, their excessive use makes the article difficult to read and understand.
Therefore, to facilitate the reading and understanding of the text of this article, I suggest that the authors minimize the use of acronyms.
Moreover, some acronyms were used, both in the body of text than in the abstract, with out being explicit. I suggest to the Authors to check it.
Moreover, the introduction result complete and well support the aim of the research and, although it is well written, there are some small typos to correct.
At line 49 “NUE” should be moved outside the square brackets.
At line 87 the acronyms “CS” was used without with out being explicit.
Again at line 87 a parenthesis should be inserted after “(DIM”.
Furthermore, the sentence from line 81 to line 86 result speculative thus they should be delete.
The paragraph of materials and methods could result, too long and some part of should need to be summarised.
For examples, the paragraph from 2.4.2 to 2.4.5 could be merged and summarized.
Moreover, the extensive use of the data reported between parenthesis, like in the text from line 111 to line 119, makes difficult to read and understand of the text.
Furthermore, data shown in table 1 and 2 and in the figure 1 should be move in the chapter result and discussion and discussed.
In addition I suggest to the Authors to check the formulas reported in the material and method since I think that the instruction for Author reported a different method to report it in the tesxt.
Finally, at line 248 the sentence “[Milkoscan 248 FT6000, Foss Analytical A/S, Hillerod, Denmark; 33]” should replaced with “(Milkoscan 248 FT6000, Foss Analytical A/S, Hillerod, Denmark)[ 33]”.
The results is sufficiently well presented and they are sufficiently discussed, also in comparison to the data reported in the literature.
However, to facilitate the understanding of the data shown in the tables by the readers it should be repot in all tables the number of the samples of each column
The conclusions result well presented and they support the aim of the research.
Author Response
General comments:
In general, acronyms can help the reader better understand the article. However, their excessive use makes the article difficult to read and understand.
Therefore, to facilitate the reading and understanding of the text of this article, I suggest that the authors minimize the use of acronyms.
AU: Acronyms now reduced significantly.
Moreover, some acronyms were used, both in the body of text than in the abstract, with out being explicit. I suggest to the Authors to check it.
AU: Checked again.
Moreover, the introduction result complete and well support the aim of the research and, although it is well written, there are some small typos to correct.
At line 49 “NUE” should be moved outside the square brackets.
AU: fixed. Line 51.
At line 87 the acronyms “CS” was used without with out being explicit.
AU: CS was defined in line 54.
Again at line 87 a parenthesis should be inserted after “(DIM”.
AU: Fixed line 86.
Furthermore, the sentence from line 81 to line 86 result speculative thus they should be delete.
AU: Now line 87-91. The authors feel that these lines introduce the topic of PTA quite well and wish to leave it in. However, if the Reviewer still feels strongly about this, we can remove. Lines 82-86.
The paragraph of materials and methods could result, too long and some part of should need to be summarised.
For examples, the paragraph from 2.4.2 to 2.4.5 could be merged and summarized.
AU: The authors wish to keep these parts separate if the Reviewer is ok with this? Sub section 2.4.2 focuses on the procedure of collecting milk samples, whilst 2.4.5 more so focuses on the timeline of how the milk samples were collected. However, if the Reviewer still feels strongly, these can be separated.
Moreover, the extensive use of the data reported between parenthesis, like in the text from line 111 to line 119, makes difficult to read and understand of the text.
AU: Much of this data in parenthesis is now removed. Lines 112-117.
Furthermore, data shown in table 1 and 2 and in the figure 1 should be move in the chapter result and discussion and discussed.
AU: I'm not sure what the Reviewer is looking for here? Is it that they want the Tables 1+ 2 and Figure 1 physically moved to another part of the manuscript or is it that they want more discussion centred around these tables and figure?
In addition I suggest to the Authors to check the formulas reported in the material and method since I think that the instruction for Author reported a different method to report it in the tesxt.
AU: All formulas have been reported as explained in instructions to authors (microsoft equation editor).
Finally, at line 248 the sentence “[Milkoscan 248 FT6000, Foss Analytical A/S, Hillerod, Denmark; 33]” should replaced with “(Milkoscan 248 FT6000, Foss Analytical A/S, Hillerod, Denmark)[ 33]”.
AU: Amended. Lines 252-253.
The results is sufficiently well presented and they are sufficiently discussed, also in comparison to the data reported in the literature.
However, to facilitate the understanding of the data shown in the tables by the readers it should be repot in all tables the number of the samples of each column
AU: Tables amended.
The conclusions result well presented and they support the aim of the research.
Reviewer 3 Report
The authors performed a feeding trial with grazing dairy cows whose supplementation with concentrates differed in CP content but not in intestinal digestibility of proteins calculated according to the INRA method. The work is impeccable as it was designed and conducted according to the classical standards of agricultural experimentation and animal science. Unfortunately, data processing and the following comments does not confirm the quality of the experimental design and conduct.
First of all, the formula for calculating the CME was applied incorrectly. If one calculates the data per column using the reported parameters, the result is quite different. For example, in the first column, the simple application of the formula results in a production of 25.88 kg of milk instead of 16.63. Furthermore, the formula is taken from grey literature, cited by https://www.sciencedirect.com/science/article/pii/S0022030216301722#bbib0220, the source of which cannot be found in the literature (R Orth - Fact Sheet A-2, Mid-states DRPC, Ames, IA, 1992). But even if the calculations were right, the ECM/kd DM ingested ratio is wrong (check to believe)
Again, in table 5 the overall crude protein concentration in the diet, g CP/kg total dry matter intake with the same data provided statistical differences, which is absurd.
Finally, the regression presented in figure 2 is meaningless and appears to be a simple software output: is the intercept different from zero? has the outlier found been evaluated? How can one comment that this relationship is good (in the text) if the Rq is very low, even if different from zero?
A note: there is no such thing as tendence to be significant! A difference is significant below a threshold (chosen by the experimenter, which may be different from the classic 5% depending on the protection of the text he intends to adopt), but to say that it tends towards that threshold is an aberration (unfortunately endured by many journals). I would suggest either removing these statements or choosing and justifying a lower level of test protection (e.g. 10%).
Given these premises, before the work can be evaluated, the authors must double-check the entire statistical analysis and verify the consistency of the data in the tables.

Author Response
The authors performed a feeding trial with grazing dairy cows whose supplementation with concentrates differed in CP content but not in intestinal digestibility of proteins calculated according to the INRA method. The work is impeccable as it was designed and conducted according to the classical standards of agricultural experimentation and animal science. Unfortunately, data processing and the following comments does not confirm the quality of the experimental design and conduct.
First of all, the formula for calculating the CME was applied incorrectly. If one calculates the data per column using the reported parameters, the result is quite different. For example, in the first column, the simple application of the formula results in a production of 25.88 kg of milk instead of 16.63. Furthermore, the formula is taken from grey literature, cited by https://www.sciencedirect.com/science/article/pii/S0022030216301722#bbib0220, the source of which cannot be found in the literature (R Orth - Fact Sheet A-2, Mid-states DRPC, Ames, IA, 1992). But even if the calculations were right, the ECM/kd DM ingested ratio is wrong (check to believe)
AU: The authors thank the Reviewer for spotting this error. The ECM parameter has now been removed from the manuscript as the authors feel that it isn't contributing to discussion.
Again, in table 5 the overall crude protein concentration in the diet, g CP/kg total dry matter intake with the same data provided statistical differences, which is absurd.
AU: Again, this parameter has been removed completely from the manuscript.
Finally, the regression presented in figure 2 is meaningless and appears to be a simple software output: is the intercept different from zero? has the outlier found been evaluated? How can one comment that this relationship is good (in the text) if the Rq is very low, even if different from zero?
AU The linear regression in Figure 2 is SAS output.
The outlier has been considered and the authors feel that although an outlier, it is not biologically impossible; hence, why we decided to leave it in.
A note: there is no such thing as tendence to be significant! A difference is significant below a threshold (chosen by the experimenter, which may be different from the classic 5% depending on the protection of the text he intends to adopt), but to say that it tends towards that threshold is an aberration (unfortunately endured by many journals). I would suggest either removing these statements or choosing and justifying a lower level of test protection (e.g. 10%).
AU: Mention of tendencies have now been completely removed from the manuscript.
Given these premises, before the work can be evaluated, the authors must double-check the entire statistical analysis and verify the consistency of the data in the tables.
AU: The authors have checked over the statistical analysis of the manuscript; hence, why we were slightly late with the resubmission. We are happy with all of the data with the exception being for the two parameters that the Reviewer brought to our attention above. These have been since removed.
Reviewer 4 Report
REVIEW
for the journal Animals (ISSN 2076-2615)
Article “Effects of protein supplementation strategy and genotype on milk production and nitrogen utilisation efficiency in late-lactation, spring-calving grazing dairy cows”
Manuscript ID: animals-1989667
Authors: Michael J. Doran, Finbar J. Mulligan, Mary B. Lynch, Alan G. Fahey, Maria Markiewicz-Keszycka, Gaurav Rajauria, Karina M. Pierce
1. The topic discussed by the authors is relevant both from a theoretical and practical point of view, since 1) there are still no studies focus on lowering the protein of the supplementary feed and how this can affect milk production and nitrogen utilization efficiency when cows of different genetic values are grazed outdoors; 2) these results are linked to the European Union's goal of reducing the negative impact of milk production on the environment.
2. In my opinion, you should explain the first abbreviation in the text of the NUE indicator (line 28).
3. Line 115. „2 cow genotypes (n = 24; Table 1;”. Parentheses are missing.
4. In my opinion, the authors should comment more on the sentence "Where no PSS × genotype interaction was observed for measured parameters, the results focus only on main effects" (lines 322–323), because readers may have questions and uncertainties about the statistical models used in the analysis and the factors used in them.
5. Taking into account the previous comment, I think it would be appropriate to present the statistical models ( y = ………..) described in the methodological part of statistical analysis (2.6 Statistical Analysis), with a precise specification of variables, fixed, random and repeated effects and interactions.
6. Also, I think it should be detailed what "REG SAS procedure" (linear regression?) was applied and whether R2 coefficients of determination (not just P values) were evaluated (line 307).
7. Under tables 3-5 there is a note: "abc - means with different superscripts differ (P < 0.05)".
The statement requires clarification: between the mean values of all indicators in the same row? Or between LCP and HCP groups?
It is also not clear why “abc” is indicated only for some indicators and not indicated for others (for example, “Milk production”, etc.).
In the "Significance" column, I propose to present all the originally obtained P values of the above-mentioned tables. Alternatively, mark all non-significant values as "non-significant". Then, in the statistical part of the methodology, the authors should clarify the "P" evaluation methodology used in the tables (P<0.05, P<0.01, P< 0.001), when "P" is significant.
8. Line 387: “We found a significant positive linear relationship (Figure 2) between total feed N intake and estimated urinary N excretion (Eq. [1], P < 0.01, R2 = 0.14)”.
As for the linear form of dependence, I strongly doubt it. Maybe the authors should pay attention to the R2 coefficient and comment on it?
9. The statement in the conclusions was unclear to me: "Genotype had no effects on milk production or cow NUE" (lines 526-527). The authors used data from the National Dairy Cattle Breeding Value System to create the groups, which is very innovative in this kind of work. I would be interested to know what the heritability of milk yield is. And I also think that the mentioned conclusion should be explained in more detail to the readers.
10. The article is interesting, but the adjustments mentioned are recommended.
Sincerely, reviewer.
Author Response
- The topic discussed by the authors is relevant both from a theoretical and practical point of view, since 1) there are still no studies focus on lowering the protein of the supplementary feed and how this can affect milk production and nitrogen utilization efficiency when cows of different genetic values are grazed outdoors; 2) these results are linked to the European Union's goal of reducing the negative impact of milk production on the environment.
- In my opinion, you should explain the first abbreviation in the text of the NUE indicator (line 28). AU: Done. Now lines 26-27.
- Line 115.„2 cow genotypes (n = 24; Table 1;”. Parentheses are missing. AU: This should be correct now. Figures between parentheses have also been cut down on to avoid confusion to the reader. Lines 114-117.
- In my opinion, the authors should comment more on the sentence "Where no PSS × genotype interaction was observed for measured parameters, the results focus only on main effects" (lines 322–323), because readers may have questions and uncertainties about the statistical models used in the analysis and the factors used in them. AU: Some more information added. Lines 335-338.
- Taking into account the previous comment, I think it would be appropriate to present the statistical models ( y = ………..) described in the methodological part of statistical analysis (2.6 Statistical Analysis), with a precise specification of variables, fixed, random and repeatedeffects and interactions. AU: Equation inserted. Lines 317-324.
- Also, I think it should be detailed what "REG SAS procedure" (linear regression?) was applied and whether R2coefficients of determination (not just P values) were evaluated (line 307). AU: Info now added. Lines 309-311.
- Under tables 3-5 there is a note: "abc - means with different superscripts differ (P < 0.05)".
The statement requires clarification: between the mean values of all indicators in the same row? Or between LCP and HCP groups? AU: Now clarified in the Tables.
It is also not clear why “abc” is indicated only for some indicators and not indicated for others (for example, “Milk production”, etc.). AU: abc is only indicated where there were significant interactions observed.
In the "Significance" column, I propose to present all the originally obtained P values of the above-mentioned tables. Alternatively, mark all non-significant values as "non-significant". Then, in the statistical part of the methodology, the authors should clarify the "P" evaluation methodology used in the tables (P<0.05, P<0.01, P< 0.001), when "P" is significant. AU: This has now been amended.
- Line 387:“We found a significant positive linear relationship (Figure 2) between total feed N intake and estimated urinary N excretion (Eq. [1], P < 0.01, R2 = 0.14)”.
As for the linear form of dependence, I strongly doubt it. Maybe the authors should pay attention to the R2 coefficient and comment on it? AU: Comment added on line 515.
- The statement in the conclusions was unclear to me: "Genotype had no effects on milk production or cow NUE" (lines 526-527). The authors used data from the National Dairy Cattle Breeding Value System to create the groups, which is very innovative in this kind of work. I would be interested to know what the heritability of milk yield is. And I also think that the mentioned conclusion should be explained in more detail to the readers. AU: Some more clarity added to the conclusion. As for the heritability of milk yield, most research reports it to be in the region of 0.15-0.25 which would mean moderate heritability. However, the authors believe that differences with respect to milk production PTA were not enough to warrant a treatment difference in our study.
- The article is interesting, but the adjustments mentioned are recommended.
Round 2
Reviewer 1 Report
Dear authors, I congratulate you on the work you have done. You made corrections to the manuscript according to my suggestions. I agree with the changes and believe the manuscript can be published as is.
Author Response
Dear authors, I congratulate you on the work you have done. You made corrections to the manuscript according to my suggestions. I agree with the changes and believe the manuscript can be published as is.
AU: The authors thank the Reviewer for their thorough review in making the manuscript stronger.
Reviewer 3 Report
Here is one of my earlier remarks and the authors' response
"Finally, the regression presented in figure 2 is meaningless and appears to be a simple software output: is the intercept different from zero? has the outlier found been evaluated? How can one comment that this relationship is good (in the text) if the Rq is very low, even if different from zero?
AU The linear regression in Figure 2 is SAS output.
The outlier has been considered and the authors feel that although an outlier, it is not biologically impossible; hence, why we decided to leave it in."
In fact, the authors did not answer this question because: 1. the fact that the regression is the output of SAS means nothing, given the law of garbage in - garbage out; 2. the intercept, given the value of the reported standard error (4 times greater than the value of the parameter) is not different from zero (and this is biologically plausible) so that the regression can be recalculated by forcing it to pass through the origin of the axes; 3. the fact that the extreme datum is biologically plausible does not mean that it belongs to the same sample population on which the statistical inference is made. I am reluctant to dump outliers without a reason, but I am equally reluctant to leave them in a data set without a reason.
The extreme data should be analysed and thoroughly and, if necessary, kept in the data set with an explanation, otherwise discarded, again with an explanation. Finally, the fact that the value of Rq is significantly different from zero does not mean that the model fits the data well; in other words, the low value of Rq found (dependent to a large extent on the extreme datum) however significantly different from zero reveals a low reliability in using the model for predictive purposes.
Author Response
"Finally, the regression presented in figure 2 is meaningless and appears to be a simple software output: is the intercept different from zero? has the outlier found been evaluated? How can one comment that this relationship is good (in the text) if the Rq is very low, even if different from zero?
AU The linear regression in Figure 2 is SAS output.
The outlier has been considered and the authors feel that although an outlier, it is not biologically impossible; hence, why we decided to leave it in."
In fact, the authors did not answer this question because: 1. the fact that the regression is the output of SAS means nothing, given the law of garbage in - garbage out; 2. the intercept, given the value of the reported standard error (4 times greater than the value of the parameter) is not different from zero (and this is biologically plausible) so that the regression can be recalculated by forcing it to pass through the origin of the axes; 3. the fact that the extreme datum is biologically plausible does not mean that it belongs to the same sample population on which the statistical inference is made. I am reluctant to dump outliers without a reason, but I am equally reluctant to leave them in a data set without a reason.
The extreme data should be analysed and thoroughly and, if necessary, kept in the data set with an explanation, otherwise discarded, again with an explanation. Finally, the fact that the value of Rq is significantly different from zero does not mean that the model fits the data well; in other words, the low value of Rq found (dependent to a large extent on the extreme datum) however significantly different from zero reveals a low reliability in using the model for predictive purposes.
AU: The intercept is different from zero. Furthermore, the authors have reconsidered the outlier. Although it is biologically possible to have this result, it it falls outside of the 95% prediction limits. Therefore we have now decided to take it out of the model. This also improves our Rr value to over 0.51, making it a more reliable as a prediction equation.